# University service utilization patterns in students with specific learning disabilities: An institutional cross-sectional study

Syed Mohammed Basheeruddin Asdaq[1,2,3]*, Mamdouh Saleh Alharbi[1], Mohanad Abdullah Alansari[1], Saleh Alshuqayr[1], Malek Fares Alanazi[1], Saleh Rajih Alshahrani[1], Naira Nayeem[4], Walaa F. Alsanie[5,6], Abdulhakeem S. Alamri[5,6], Majid Alhomrani[5,6], Amal F. Alshammary[7], Rafiulla Gilkaramenthi[8]

**1** Department of Pharmacy Practice, College of Pharmacy, AlMaarefa University, Ad Diriyah, Saudi Arabia, **2** King Salman Center for Disability Research, Riyadh, Saudi Arabia, **3** Research Center, Deanship of Scientific Research and Post-Graduate Studies, AlMaarefa University, Dariyah, Riyadh, Saudi Arabia, **4** Department of Pharmaceutical Chemistry, Faculty of Pharmacy, Northern Border University, Rafha, Saudi Arabia, **5** Department of Clinical Laboratory Sciences, The Faculty of Applied Medical Sciences, Taif University, Taif, Saudi Arabia, **6** Research Center for Health Sciences, Deanship of Graduate Studies and Scientific Research, Taif University, Taif, Saudi Arabia, **7** Department of Clinical Laboratory Sciences, College of Applied Medical Sciences, King Saud University, Riyadh, Saudi Arabia, **8** Department of Emergency Medical Services, College of Applied Sciences, AlMaarefa University, Diriyah, Riyadh, Saudi Arabia

* sasdaq@gmail.com, sasdag@um.edu.sa

## Abstract

### Background/Objectives

This study examined the relationship between university students' use of common and specific services and the presence of specific learning disabilities (SLDs). The objective was to identify how sociodemographic factors, health status, and learning disabilities impact service utilization, with a focus on improving accessibility and support for students with SLDs.

### Methods

Employing random but voluntary sampling, a cross-sectional survey of university students was conducted using a tool demonstrating good internal reliability (Cronbach's α = 0.859). The survey collected data on sociodemographics, health status, and self-reported specific learning disabilities (SLDs). Participants assessed their utilization of common university services (e.g., library, academic advising, LMS) and specific support services (e.g., one-to-one meetings, electronic books, dispensatory measures, compensatory tools). Data analysis included frequency distributions, Chi-square tests, Binary Logistic Regression, and Pearson correlation to examine associations between SLDs and service utilization patterns.

**Data availability statement:** All relevant data are within the manuscript and its Supporting Information files (Supplementary file 1, 2 and 3)

**Funding:** The authors extend their appreciation to the King Salman center For Disability Research for funding this work through Research Group no KSRG-2024-027.

**Competing interests:** The authors have declared that no competing interests exist.

## Results

Participants (n = 302) were predominantly male (59.9%) and aged 18–25 years (89.7%), with high rates of dysgraphia (42.7%), dyscalculia (43%), and auditory processing disorder (23.8%). Regression analyses revealed: Frequent use of common services was significantly associated with visual perceptual/motor deficits (aOR=3.87, 95% CI = 1.82–8.21, $P < .001$), advanced academic year (aOR=1.29, 95% CI = 1.09–1.53, $P = 0.004$), and positive educational self-perception (aOR=2.32, 95% CI = 1.12–4.80, $P = 0.024$). For specific services, key predictors included female gender (aOR=2.07, 95% CI = 1.15–3.73, $P = 0.015$), dyslexia (aOR=2.73, 95% CI = 1.07–6.98, $P = 0.036$), auditory processing disorder (aOR=2.52, 95% CI = 1.17–5.41, $P = 0.018$), while sleep disturbances reduced utilization (aOR=0.46, 95% CI = 0.27–0.79, $P = 0.005$).

## Conclusion

This study reveals significant underutilization of university services among students with specific learning disabilities (SLDs), despite high prevalence rates. Engagement patterns were strongly influenced by SLD type, academic progression, self-perception, gender, and health factors. To address these barriers, we recommend targeted interventions including: 1) disability-awareness campaigns to reduce stigma, 2) tailored accommodation (e.g., extended time, multimodal materials) aligned with specific SLD profiles, and 3) mandatory faculty training on inclusive instructional strategies and available support resources. These evidence-based measures are critical for improving accessibility and academic success for this vulnerable population.

## Introduction

Specific Learning Disabilities (SLDs) encompass a range of disorders characterized by significant challenges in acquiring and applying skills such as listening, speaking, reading, writing, reasoning, or mathematical abilities [1]. These challenges are intrinsic to the individual and are thought to arise from dysfunctions within the central nervous system (CNS) [2,3]. Unlike other types of disabilities, SLDs are primarily related to learning and education. As learning is a cognitive process, any disruption to brain function can significantly hinder it.

In the U.S., SLDs are the most common disability reported by postsecondary students [4]. Among all students with disabilities at the postsecondary level, 31% have learning disabilities [5]. Over the past few decades, the representation of students with disabilities in U.S. higher education has increased, largely due to federal policies such as the Individuals with Disabilities Education Act (IDEA) of 1990, its 1997 amendments, and its 2004 reauthorization, which promoted greater inclusion and access to educational opportunities [6]. Students today in postsecondary education should benefit from the secondary-to-postsecondary transition services mandated by IDEA 2004.

Globally, there has been an increased focus on improving the accessibility of higher education, including in research, policy, and practice. Despite this progress, students with disabilities continue to face significant barriers in learning and assessment [7,8] and are systematically disadvantaged at every stage of their academic journey. Research shows that students with disabilities, including those with learning disabilities, generally experience lower academic performance and retention rates compared to their peers without disabilities [9]. Although disabled students are entitled to various accommodations and support services, typically coordinated by a Centre of Disability Services, the lack of these essential services increases the risk of dropout. Studies have highlighted that nearly 70% of college students with learning disabilities drop out compared to their non-disabled peers, with these students tending to earn lower GPAs, take more leaves of absence, and switch to less demanding programs that prepare them for less rewarding careers [10,11].

In response to the multifaceted academic challenges faced by students with SLDs, universities across various countries have increasingly implemented specialized support services aimed at promoting inclusive education and enhancing student success. These learning disability services typically offer a range of academic accommodations and support interventions, including but not limited to individualized or group tutoring, academic coaching, mental health counseling, and access to psychological assessments. Moreover, assistive technologies and learning aids form a critical component of these services, enabling students to overcome barriers related to reading, writing, time management, and organization. Despite the growing recognition of the importance of such support systems, there remains a paucity of systematic and longitudinal research exploring the effectiveness and utilization of accommodations and broader university-provided services for students with SLDs [12]. Understanding which services are most beneficial, and how they are accessed and perceived by students, is essential for developing evidence-based institutional policies and enhancing support structures. A study by Sumner et al. [13] provided valuable insights by comparing the accessibility and utilization of various specialized support services among students diagnosed with dyslexia and developmental coordination disorder (DCD). Their findings highlighted that technology-based interventions—such as the use of laptops, text-to-speech software, digital mind-mapping tools, and audio-recording devices—were the most accessed resources. In contrast, direct academic support in the form of one-on-one or group tutoring sessions was comparatively less frequently utilized, potentially due to resource constraints, student preferences, or lack of awareness regarding service availability. Recent studies corroborate these findings, emphasizing the increasing reliance on digital technologies to support students with learning difficulties, particularly in higher education environments. For instance, MacCullagh et al. [14] found that assistive technology significantly enhances the academic performance and independence of students with dyslexia when integrated effectively into learning environments. Similarly, Mull & Sitlington [15] argue that postsecondary institutions must adopt comprehensive transition planning and adaptive support systems to ensure academic persistence and reduce dropout rates among students with disabilities.

Although SLDs are well-documented in Western educational contexts with prevalence rates of 5–15% [12], they represent a relatively recent focus in Saudi Arabia. The country officially recognized learning disabilities as a special education category only in 1996 [16], enabling formal intervention programs and identification methodologies. Despite this critical development, there remains a significant lack of understanding regarding how SLDs manifest and affect academic experiences in the Arab world, particularly in Saudi Arabia. Furthermore, research examining how sociodemographic factors—including age, gender, nationality, and family income—interact with university service utilization is notably limited. Compounding this gap, cultural and societal factors such as family expectations and disability stigma may profoundly influence how Saudi students with SLDs perceive their academic abilities and satisfaction [17]. These intersecting elements shape academic experiences in ways not yet adequately explored.

While Saudi universities have established support services in recent decades, comprehensive data on SLD-specific prevalence and service efficacy remain scarce. This study, therefore, investigates the relationship between sociodemographic factors, SLDs, and utilization of both general and specialized university services among students in Saudi Arabia—addressing critical regional knowledge gaps while accounting for unique cultural dimensions.

## Subjects and methods

### Study design

This was a cross-sectional, observational study aimed at assessing the use of both common and specific services offered to students at AlMaarefa University. To evaluate the use of services across the students' community and to compare based on the presence of SLDs, the questionnaire was distributed to all prospective participants, allowing for the collection of data on both general and specialized services. A cross-sectional design was chosen to efficiently capture data at a single point in time, providing a snapshot of associations between sociodemographic factors, learning disabilities, and the use of both types of services, as well as their correlation with students' health status.

### Study location

The study was conducted at AlMaarefa University in Riyadh, Saudi Arabia, which offers programs in clinical pharmacy, computer science/information science, and allied health sciences (nursing, respiratory therapy, anesthesia technology, and emergency care). Medical programs were excluded from the study due to their different educational requirements. The study setting was ideal for examining how students with specific learning disabilities use services that might impact their learning outcomes. The study was conducted from April to June 2024.

### Questionnaire contents

A self-administered questionnaire was designed to collect data on sociodemographic characteristics, health status, self-reported learning disabilities, and the use of common and specific university services (S1 File). The questionnaire was divided into the following sections:

- **Sociodemographic Information**: This section captured information on students' gender, age, nationality, program, academic year, Cumulative Grade Point Average (cGPA), and monthly family income (SAR).

- **Health Status**: This section assessed students' self-reported health status, including body mass index (BMI), quality of life (QOL), satisfaction with health, chronic diseases, sleep duration, and physical activity levels.

- **Specific Learning Disabilities (SLDs)**: Students self-reported whether they had ever been diagnosed with or experienced symptoms of any learning disabilities, such as dyslexia, dysgraphia, dyscalculia, auditory processing disorder, language processing disorder, nonverbal learning disabilities, and visual perceptual deficits. No clinical verification was required, as our objective was to capture students' self-perceived learning challenges regardless of formal diagnosis status. The reliability of this subscale was confirmed with a Cronbach's alpha value of 0.793.

- **Use of Common University Services**: This section evaluated students' use of various university services, including the library, academic portal for registration, Moodle, academic databases, attendance in lectures, orientation sessions, academic advising, psychological assistance, university apps, canteen facilities, and sports/fitness services. Students rated their usage frequency on a 5-point scale: 1 = Very rarely, 2 = Rarely, 3 = Sometimes, 4 = Often, and 5 = Very often. Internal consistency for this scale was strong (α = 0.835).

- **Use of Specific University Services**: This section assessed services for students with specific learning difficulties, such as academic support (one-on-one tutoring, electronic books), academic counseling, psychological support, compensatory measures (e.g., splitting lecture material, extra time for exams), and technology tools (e.g., digital recorders, reading pens). Responses were recorded on a 5-point agreement scale: 1 = Strongly disagree, 2 = Disagree, 3 = Neutral, 4 = Agree, and 5 = Strongly agree. Scale reliability was good (α = 0.820).

The questionnaire was pilot-tested with 20 participants randomly selected from the eligible student population to evaluate clarity, feasibility, and item comprehension. Feedback was incorporated to refine wording and structure, resulting in an

instrument demonstrating excellent internal consistency ($\alpha = 0.859$). Data from this preliminary phase were not included in the final analysis.

## Study samples

The required sample size was calculated based on the population of approximately 1,700 students in the pharmacy, allied health, and computer science programs. A sample size of 115 students was initially calculated with a 95% confidence level and 5% margin of error, based on an 8.76% prevalence of specific learning disabilities [18] (http://www.raosoft.com/samplesize.html). However, a power analysis confirmed that a final sample of 302 students would provide sufficient statistical power (80%) to detect meaningful relationships among the variables, including the use of university services and specific learning disabilities.

## Eligibility criteria

Inclusion Criteria: Students from AlMaarefa University, aged ≥18 years, enrolled in programs such as PharmD, Nursing, Anesthesia Technology, Respiratory Care, Emergency Care, Computer Science, and Information Sciences. Students from all academic years, including first-year through internship year, were eligible. Participation was voluntary, and informed consent was obtained from all participants.

Exclusion Criteria: Students enrolled in medical programs or who did not provide informed consent were excluded. Additionally, students who did not complete the questionnaire or provided incomplete responses were excluded from the analysis.

## Ethical considerations

The study was approved by the Institutional Review Board (IRB) of AlMaarefa University (IRB24−21) and adhered to STROBE guidelines for observational research. Oral informed consent was obtained from all participants following a scripted disclosure process approved by the IRB. This process included a verbal explanation of the study's purpose, hypothesis, expected outcomes, risks, benefits, and the voluntary nature of participation. Participants were assured of the confidentiality and anonymity of their responses and their right to withdraw at any time without penalty. Participants accessed the online questionnaire via QR code only after confirming their consent verbally. The study's purpose and consent terms were reiterated at the beginning of the questionnaire, and only those who agreed proceeded with the survey. The IRB waived the requirement for written consent due to the minimal risk nature of the study and to protect participant anonymity, but all oral consent procedures were explicitly approved by the IRB protocol (IRB24−21). Ethical standards were strictly maintained throughout the study.

## Sample recruitment

Participants were randomly selected from the eligible student population (N ≈ 1,700) using institutional enrollment lists. Invitations were sent to 340 randomly chosen students via email and university portals. Of these, 302 completed the study (89% retention), while 38 withdrew during data collection. Although selection was random, participation was voluntary. Thus, our final sample may reflect self-selection bias (e.g., students with stronger opinions about services or greater service exposure being more likely to respond). Demographic comparisons with institutional data confirmed broad representativeness across academic years but revealed slight underrepresentation of first, third, and final year students, compared to second, fourth, and fifth years.

## Statistical analysis

Data were analyzed using SPSS version 25 (IBM, Armonk, NY). Descriptive statistics summarized sociodemographic characteristics, health status, and use of common and specific services. Chi-square tests were used to compare proportions of service usage across different sociodemographic and health status categories. Binary Regression analysis was performed

to identify factors influencing the use of university services, adjusting for sociodemographic variables and the presence of learning disabilities. Pearson correlation analysis assessed relationships between specific learning disabilities and service usage. A *P* value <0.05 was considered statistically significant, and results were reported with 95% confidence intervals.

## Results

### Sociodemographic characteristics of participants

Most of the participants were male (59.9%), aged 18–25 years (89.7%), and Saudi nationals (64.2%) (Fig 1). Most were enrolled in the Pharm.D program (53%) and in the later years of their academic programs, with the largest group being in their first year (23.5%). Regarding academic performance, 49% had a cGPA of 3.5 or higher, and 70.5% reported good educational outcomes. In terms of family income, 70.5% had a total income of 10,000 SAR or more.

### Health status of the participants

A large majority of participants rated their health as good (97.4%) and reported no chronic diseases (80.8%) (Fig 2). However, more participants had below-normal sleep status (55.3%) than normal sleep (44.7%). Over half of the participants were physically active (57.3%). In terms of BMI, the majority had a healthy weight (47.4%), while 29.1% were overweight, 13.2% were obese, and 10.3% were underweight. Regarding quality of life, 64.6% declared it as good, and 60.3% were satisfied with their health, although a significant portion reported below-average satisfaction and quality of life.

### Status of specific learning disabilities

Many participants were not affected by dyslexia (84.8%), dysgraphia (57.3%), dyscalculia (57.0%), auditory processing disorder (76.2%), language processing disorder (67.9%), nonverbal learning disabilities (69.2%), and visual perceptual/visual motor deficits (71.9%) (Fig 3). However, notable percentages of participants reported the presence of these disabilities, with

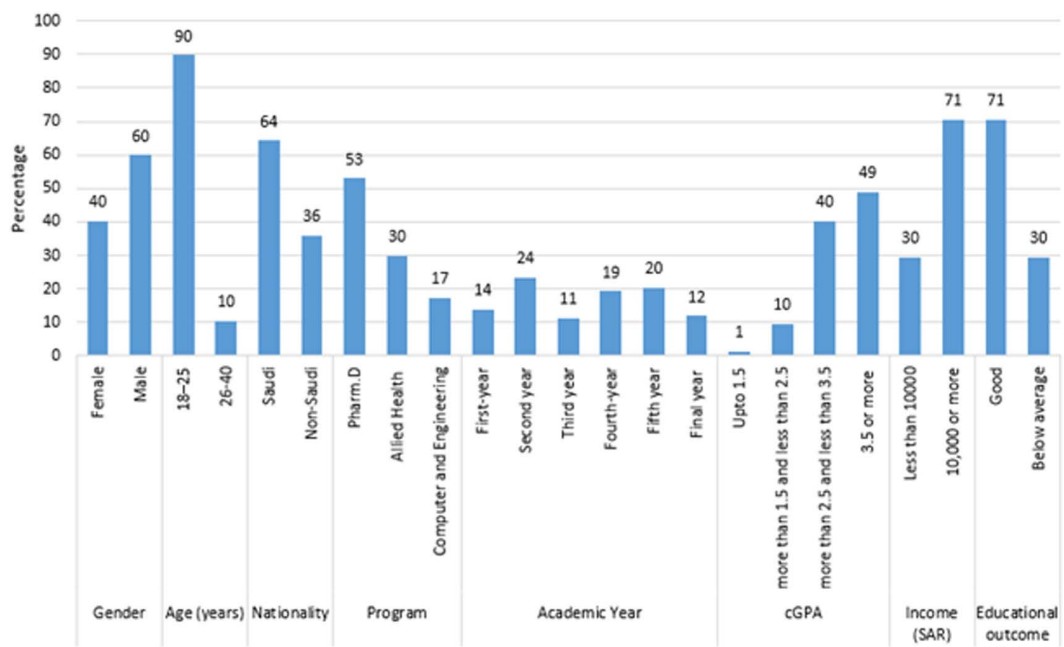

**Fig 1. Sociodemographic characteristics of the participants.**

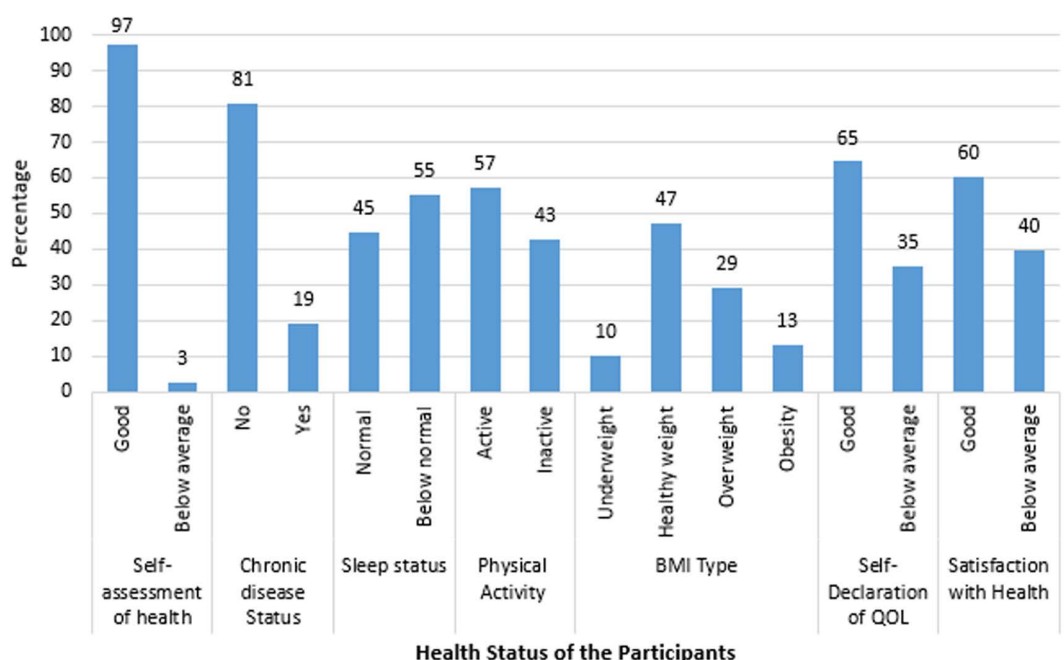

**Fig 2. Health status of the participants.**

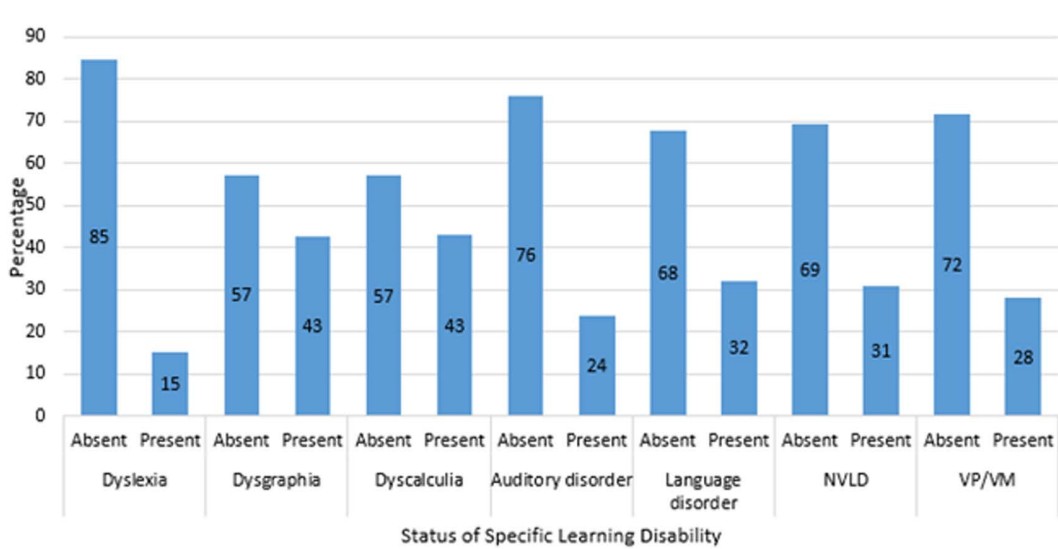

**Fig 3. Frequency distribution of specific learning disabilities (SLD).**

dysgraphia (42.7%), dyscalculia (43.0%), auditory processing disorder (23.8%), language processing disorder (32.1%), nonverbal learning disabilities (30.8%), and visual perceptual/visual motor deficits (28.1%) being the most prevalent.

## Use of university services

The data in Fig 4 shows that 44.7% of participants reported using common university facilities often, while a slightly higher percentage, 55.3%, indicated that they use these facilities rarely. Further, core academic activities dominate usage.

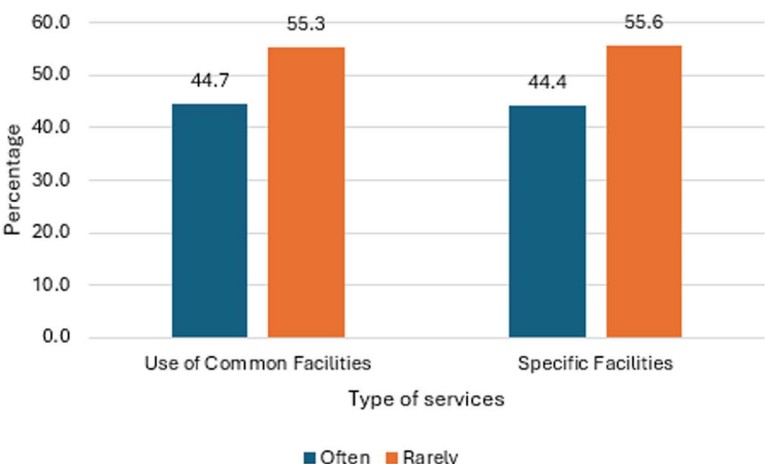

**Fig 4. Status of use of services by participants.**

Lectures (3.75) and the Learning Management System (LMS) (3.72) are the most heavily utilized services, reflecting their fundamental role in course delivery and student workflow. Library services (3.38) also see relatively high engagement (Fig 5). Moderate usage is observed for academic advising (2.61), orientations (2.56), specialized academic resources like Databases (2.58) and Lesson Plans/Study Materials (2.96), and the Canteen (2.23). Significantly lower usage characterizes support and wellness services: Psychological Counseling (1.55) and Fitness/Sports facilities (1.87) are the least used, despite their importance for student well-being. Academic Portal usage (2.01) also falls into this lower tier.

The usage of specific services were reported by 44.4%, while 55.6% reported rare utilization of those services among the participants (Fig 4). Utilization of specific university services (rated 1–5) reveals distinct patterns. Both one-to-one

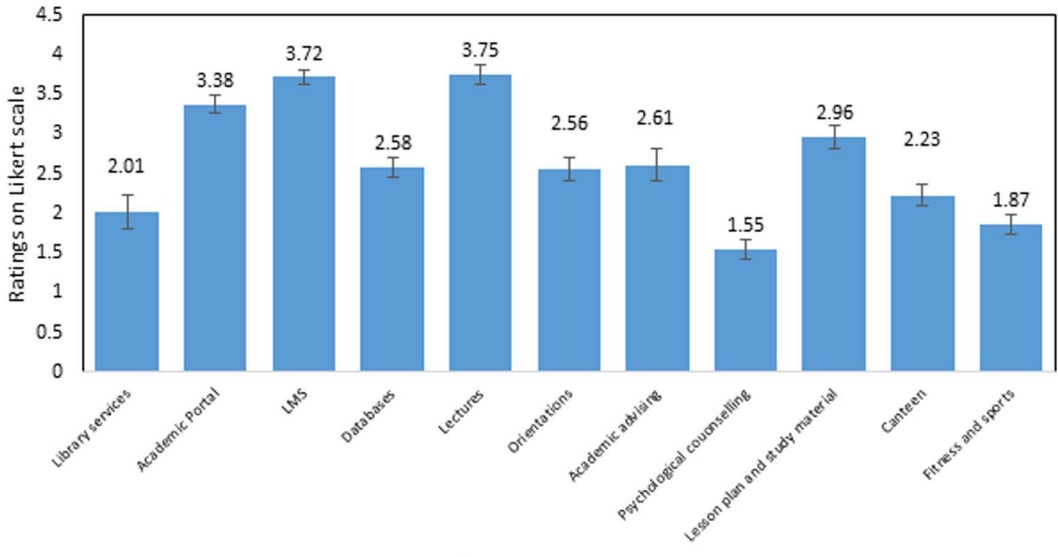

**Fig 5. Usage pattern of common university services.** Data given in mean±SEM; LMS: Learning management system; Responses measured on a 5-point Likert scale.

meetings and use of electronic books show the highest engagement (3.16), indicating strong student adoption of personalized support and digital resources. Use of dispensatory measures (2.87) and compensatory tools (2.82) demonstrate moderate but consistent usage. In stark contrast, Psychological Consultation reports very low adoption (1.35), highlighting a significant gap in mental health service utilization ([Fig 6]).

### Associations between sociodemographic characteristics and utilization of common and specific services

[Table 1] compares the use of common services among participants based on various sociodemographic characteristics. Gender analysis reveals no significant difference in the use of common services, as both females (44%) and males (45%) report similar usage rates ($P=0.797$). Age does not show a notable difference in service use, with both the 18–25 and 26–40 age groups using common services at comparable rates ($P=0.663$). Nationality also has little effect, as both Saudi (43%) and non-Saudi (48%) participants report similar usage patterns ($P=0.369$). Academic program affiliation reveals that Allied Health students use common services more frequently (53%) compared to Pharm.D students (45%) and students from other programs ($P=0.018$). Academic year shows that first-year students use common services more often (62%), while usage decreases in later years, particularly for third-year students (35%) and fourth-year students (31%) ($P=0.030$). cGPA and family income show no significant effect on the use of common services, though slight trends suggest higher cGPA and income levels correlate with more frequent use. Finally, self-declared educational outcome indicates that students who rate their performance as "good" use common services more frequently (48%) than those who rate their performance as "below average" (36%) ($P=0.048$).

For specific services, gender differences are significant, with females (54%) using specific services more often than males (38%) ($P=0.008$). Age group differences also emerge, but they are not statistically significant ($P=0.216$). Regarding nationality, non-Saudis (52%) use specific services more frequently than Saudis (40%), but this difference is only marginally significant ($P=0.051$). Academic program affiliation shows no significant difference in the use of specific services across the various programs ($P=0.428$). Academic year has a stronger influence on the use of specific services, as first-year students report the lowest usage (24%), while third-year students use specific services the most (75%) ($P=0.001$). cGPA and family income do not show significant effects on specific service use, although a slight trend indicates higher cGPA correlates with more frequent usage. Lastly, self-declared educational outcome demonstrates a strong association, with students rating their performance as "good" using specific services more often (51%) compared to those with a "below average" rating (29%) ($P=0.001$).

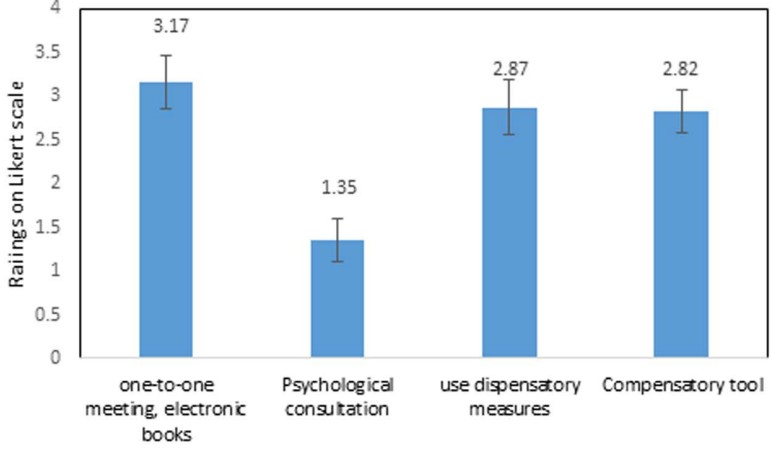

**Fig 6. Usage pattern of specific university services.** Data given in mean±SEM; Responses measured on a 5-point Likert scale.

**Table 1. Associations of sociodemographic characteristics with utilization of common and specific services.**

| # | Characteristics | Variables | Common services, n (%) | | P value* | Specific services, n (%) | | P value* |
|---|---|---|---|---|---|---|---|---|
| | | | Often | Rarely | | Often | Rarely | |
| 1. | Gender | Female | 53 (44) | 68 (56) | 0.797 | 65 (54) | 56 (46) | **0.008** |
| | | Male | 82 (45) | 99 (55) | | 69 (38) | 112 (62) | |
| 2. | Age (years) | 18–25 | 120 (44) | 151 (56) | 0.663 | 117 (43) | 154 (57) | 0.216 |
| | | 26-40 | 15 (48) | 16 (52) | | 17 (55) | 14 (45) | |
| 3. | Nationality | Saudi | 83 (43) | 111 (57) | 0.369 | 78 (40) | 116 (60) | 0.051 |
| | | Non-Saudi | 52 (48) | 56 (52) | | 56 (52) | 52 (48) | |
| 4. | Affiliated Program | Pharm.D | 72 (45) | 88 (55) | 0.018 | 75 (47) | 85 (53) | 0.428 |
| | | Allied Health | 48 (53) | 42 (47) | | 40 (44) | 50 (56) | |
| | | Computer and Engineering | 15 (29) | 37 (71) | | 19 (37) | 33 (63) | |
| 5. | Academic Year | First-year | 26 (62) | 16 (38) | **0.030** | 10 (24) | 32 (76) | **0.001** |
| | | Second year | 36 (51) | 35 (49) | | 30 (42) | 41 (58) | |
| | | Third year | 12 (35) | 22 (65) | | 25 (75) | 9 (25) | |
| | | Fourth year | 18 (31) | 40 (69) | | 24 (41) | 34 (59) | |
| | | Fifth year | 25 (41) | 36 (59) | | 25 (41) | 36 (59) | |
| | | Final year | 18 (50) | 18 (50) | | 20 (56) | 16 (44) | |
| 7. | cGPA[a] | Upto 1.5 | 1 (25) | 3 (75) | 0.474 | 3 (75) | 1 (25) | 0.522 |
| | | more than 1.5 and less than 2.5 | 16 (55) | 13 (45) | | 11 (38) | 18 (62) | |
| | | more than 2.5 and less than 3.5 | 56 (46) | 65 (54) | | 52 (43) | 69 (57) | |
| | | 3.5 or more | 62 (42) | 86 (58) | | 68 (46) | 80 (54) | |
| 8. | Total family income (SAR)[$] | Less than 10000 | 35 (39) | 54 (61) | 0.224 | 37 (42) | 52 (58) | 0.527 |
| | | 10,000 or more | 100 (47) | 113 (53) | | 97 (46) | 116 (54) | |
| 9. | Self-declaration of educational outcome | Good | 103 (48) | 110 (52) | **0.048** | 108 (51) | 105 (49) | **0.001** |
| | | Below average | 32 (36) | 57 (64) | | 26 (29) | 63 (71) | |

[a]cGPA: Cumulative Grade Point Average; [$]SAR: Saudi Arabian Riyal; *P value: Chi-Square Test; P value < 0.05 indicates a statistically significant difference between sociodemographic characteristics and outcome variables; Bolded P values indicate significance.

## Comparisons of the use of common and specific services with health status

Table 2 compares the use of common services among participants based on their health status. In terms of self-assessed health, there is no significant difference in the frequency of common service use between those who rated their health as "Good" and those who rated it as "Below average" ($P = 0.081$). Regarding chronic disease status, individuals without chronic diseases use common services similarly to those with chronic diseases, with no significant difference ($P = 0.542$). Sleep status shows a significant difference, as participants with "Normal" sleep patterns use common services more often (51%) compared to those with "below normal" sleep patterns (40%) ($P = 0.044$). Physical activity status also plays a role, with active individuals using common services more frequently (51%) than inactive individuals (36%) ($P = 0.006$). Body mass index (BMI) shows a slight trend where underweight and obese participants use common services more often, but no significant differences are observed across BMI categories ($P = 0.053$). Self-declared quality of life (QOL) and satisfaction with health both show no significant differences in common service use, although individuals reporting a "Good" QOL or health satisfaction tend to use common services more often than those with "Below average" ratings ($P = 0.158$ and $P = 0.182$, respectively).

For specific services, self-assessed health status does not significantly influence usage, as both "Good" and "Below average" health status groups report similar frequencies of use ($P = 0.692$). Chronic disease status, however, shows a

**Table 2. Comparisons of the use of common and specific services with health status.**

| # | Characteristics | Variables | Common services, n (%) | | P value* | Specific services, n (%) | | P value* |
|---|---|---|---|---|---|---|---|---|
| | | | Often | Rarely | | Often | Rarely | |
| 1. | Self-assessment of health | Good | 129 (44) | 165 (56) | 0.081 | 131 (45) | 163 (55) | 0.692 |
| | | Below average | 6 (75) | 2 (25) | | 3 (37) | 5 (63) | |
| 2. | Chronic disease Status | No | 107 (44) | 137 (56) | 0.542 | 115 (47) | 129 (53) | **0.048** |
| | | Yes | 28 (48) | 30 (52) | | 19 (33) | 39 (67) | |
| 3. | Sleep status | Normal | 69 (51) | 66 (49) | **0.044** | 49 (36) | 86 (64) | 0.011 |
| | | Below normal | 66 (40) | 101 (60) | | 85 (51) | 82 (49) | |
| 4. | Physical Activity | Active | 89 (51) | 84 (49) | 0.006 | 80 (46) | 93 (54) | 0.448 |
| | | Inactive | 46 (36) | 83 (64) | | 54 (42) | 75 (58) | |
| 5. | BMI Type[a] | Underweight | 16 (52) | 15 (48) | 0.053 | 13 (42) | 18 (58) | 0.808 |
| | | Healthy weight | 68 (48) | 75 (52) | | 60 (42) | 83 (58) | |
| | | Overweight | 29 (33) | 59 (67) | | 42(48) | 46 (52) | |
| | | Obesity | 22 (55) | 18 (45) | | 19 (48) | 21 (52) | |
| 6. | Self-Declaration of QOL[#] | Good | 93 (48) | 102 (52) | 0.158 | 94 (48) | 101 (52) | 0.070 |
| | | Below average | 42 (39) | 65 (61) | | 40 (37) | 67 (63) | |
| 7. | Satisfaction with Health | Good | 87 (48) | 95 (52) | 0.182 | 80 (44) | 102 (56) | 0.858 |
| | | Below average | 48 (40) | 72 (60) | | 54 (45) | 66 (55) | |

[#]QOL: Quality of life; [a]BMI: Body Mass Index; *P value: Chi-Square Test; P value<0.05 indicates a statistically significant difference between health status and outcome variables; Bolded P values indicate significance.

significant difference, with participants without chronic diseases using specific services more frequently (47%) compared to those with chronic diseases (33%) (P=0.048). Sleep status has a notable impact, with those reporting "Below normal" sleep patterns using specific services more often (51%) than those with "Normal" sleep patterns (36%) (P=0.011). Physical activity status does not significantly affect the use of specific services, as active and inactive individuals report similar usage rates (P=0.448). BMI categories do not show significant differences in specific service use, with individuals in all weight categories using specific services at comparable rates (P=0.808). Both self-declared QOL and satisfaction with health indicate no significant differences in the use of specific services, although individuals with a "Good" QOL or satisfaction tend to use specific services slightly more often than those with a "Below average" rating (P=0.070 and P=0.858, respectively).

## Comparisons of the use of common and specific services with specific learning disabilities

Table 3 compares the use of common and specific services among participants with and without specific learning disabilities. For common services, the presence of certain learning disabilities significantly affects usage. Dyslexia does not show significant difference in the use of common services, with 47% of individuals without dyslexia using common services often compared to 35% of those with dyslexia (P=0.142). However, participants with dysgraphia use common services less frequently (36%) compared to those without it (51%) (P=0.013). The presence of dyscalculia and auditory processing disorder does not significantly influence common service use, as both groups report similar usage rates (P=0.153 and P=0.159, respectively). Language processing disorder shows a significant difference, with individuals affected by this disorder using common services less frequently (33%) compared to those without it (50%) (P=0.005). The presence of nonverbal learning disabilities does not significantly affect the use of common services (P=0.099). Finally, individuals with visual perceptual/visual motor deficits use common services significantly less often (25%) compared to those without it (53%) (P=0.001).

**Table 3. Comparison of the use of common and specific services with SLDSs.**

| Number | Characteristics | Variables | Common services, n (%) | | P value* | Specific services, n (%) | | P value* |
|---|---|---|---|---|---|---|---|---|
| | | | Often | Rarely | | Often | Rarely | |
| 1. | Dyslexia | Absent | 119 (47) | 137 (53) | 0.142 | 124 (48) | 132 (52) | **0.001** |
| | | Present | 16 (35) | 30 (65) | | 10 (22) | 36 (78) | |
| 2. | Dysgraphia | Absent | 88 (51) | 85 (49) | **0.013** | 85 (49) | 88 (51) | 0.054 |
| | | Present | 47 (36) | 82 (64) | | 49 (38) | 80 (62) | |
| 3. | Dyscalculia | Absent | 83 (48) | 89 (52) | 0.153 | 81 (47) | 91 (53) | 0.273 |
| | | Present | 52 (40) | 78 (60) | | 53 (41) | 77 (59) | |
| 4. | Auditory processing disorder | Absent | 108 (47) | 122 (53) | 0.159 | 114 (50) | 116 (50) | **0.001** |
| | | Present | 27 (38) | 45 (63) | | 20 (28) | 52 (72) | |
| 5. | Language processing disorder | Absent | 103 (50) | 102 (50) | **0.005** | 99 (48) | 106 (52) | **0.046** |
| | | Present | 32 (33) | 65 (67) | | 35 (36) | 62 (64) | |
| 6. | Nonverbal learning disabilities | Absent | 100 (48) | 109 (52) | 0.099 | 99 (47) | 110 (53) | 0.116 |
| | | Present | 35 (38) | 58 (62) | | 35 (38) | 58 (62) | |
| 7. | Visual perceptual/visual motor deficit | Absent | 114 (53) | 103 (48) | **0.001** | 101 (47) | 116 (53) | 0.225 |
| | | Present | 21 (25) | 64 (75) | | 33 (39) | 52 (61) | |

*P value: Chi-Square Test; P value < 0.05 indicates a statistically significant difference between specific learning disabilities and outcome variables; Bolded P values indicate significance.

For specific services, the presence of specific learning disabilities has a more pronounced impact. Dyslexia shows a significant difference, with those affected using specific services less frequently (22%) compared to those without it (48%) ($P = 0.001$). The use of specific services for dysgraphia is not significantly different, though individuals with dysgraphia report a slightly higher frequency of use (38%) compared to those without it (49%) ($P = 0.054$). Similarly, the presence of dyscalculia does not significantly affect the use of specific services ($P = 0.273$), with both groups using services at comparable rates. Auditory processing disorder has a significant effect, as those with the disorder use specific services less frequently (28%) compared to those without it (50%) ($P = 0.001$). Language processing disorder also shows a significant difference, with affected individuals using specific services less often (36%) compared to those without it (48%) ($P = 0.046$). There are no significant differences in the use of specific services for participants with non-verbal learning disabilities ($P = 0.116$). Finally, while visual perceptual/visual motor deficit does not significantly affect the use of specific services ($P = 0.225$), individuals with this condition use services less frequently (39%) compared to those without it (47%).

The correlations between specific learning disabilities (SLDs) and types of common university service usage based on Pearson Chi-Square $P$-values ($P < 0.05$ indicates statistical significance) are given in Table 4. Analysis reveals distinct patterns of university service usage significantly correlated with specific learning disabilities. Academic advising, lectures, and lesson plans/study materials are critical across nearly all SLDs (Dyslexia, Dysgraphia, Dyscalculia, Auditory Processing Disorder (APD), Language Processing Disorder (LPD), Nonverbal Learning Disabilities (NVLD), and Visual Perceptual/Motor Deficit), highlighting their foundational role. Orientations also show broad significance. Digital resources (LMS, Databases, Academic Portal) are particularly vital for SLDs involving processing challenges (APD, NVLD, Visual Deficit, LPD, Dyscalculia). Wellness services exhibit targeted relevance: Canteen use correlates with Dyslexia, Dysgraphia, and Visual Deficit, while Fitness/Sports links to Dyslexia, NVLD, and Visual Deficit. Library services are uniquely associated with APD. Notably, Psychological Counseling shows no significant correlation with any SLD, aligning with its overall low utilization. These patterns emphasize the need for tailored academic and structural supports, while underscoring a gap in mental health service engagement among students with learning disabilities.

**Table 4. Utilization of types of common university services and their association with SLDs.**

| Specific Learning Disabilities | Types of Common University Services | | | | | | | | | | |
|---|---|---|---|---|---|---|---|---|---|---|---|
| | Library services | Academic Portal | LMS | Databases | Lectures | Orientations | Academic advising | Psychological counselling | Lesson plan and study material | Canteen | Fitness and sports |
| Dyslexia | 0.594 | **0.019** | 0.068 | 0.067 | **0.001** | **0.001** | **0.001** | 0.426 | **0.001** | **0.017** | **0.005** |
| Dysgraphia | 0.554 | 0.638 | 0.545 | 0.048 | 0.458 | **0.016** | **0.008** | 0.569 | 0.058 | **0.022** | 0.389 |
| Dyscalculia | 0.123 | **0.015** | 0.124 | 0.739 | **0.001** | **0.028** | 0.189 | 0.772 | **0.030** | 0.532 | 0.096 |
| Auditory processing disorder | **0.037** | **0.007** | **0.001** | **0.015** | **0.001** | **0.015** | **0.048** | 0.611 | 0.191 | 0.891 | 0.30 |
| Language processing disorder | **0.09** | **0.035** | 0.063 | 0.079 | 0.153 | **0.001** | **0.001** | 0.75 | **0.001** | 0.431 | 0.102 |
| Nonverbal learning disabilities | 0.128 | **0.008** | **0.012** | **0.001** | **0.016** | 0.052 | **0.001** | 0.258 | **0.001** | 0.594 | **0.003** |
| Visual perceptual/visual motor deficit | 0.085 | **0.011** | 0.056 | **0.001** | **0.002** | 0.069 | **0.001** | 0.844 | **0.001** | 0.012 | **0.006** |

Bolded *P* values indicate significance.

Based on Pearson Chi-Square p-values (*P* < 0.05 indicates statistical significance), the analysis reveals significant correlations between specific learning disabilities (SLDs) and the utilization of specific university support services (Table 5). One-to-one meetings show highly significant correlations with nearly all SLDs (Dyslexia *P* = 0.001, Dysgraphia *P* = 0.007, Auditory Processing Disorder *P* = 0.019, Language Processing Disorder *P* = 0.001, Nonverbal LD *P* = 0.002, Visual Motor Deficit *P* = 0.017), indicating these personalized sessions are critical for diverse learning needs. Electronic books are significantly associated with Dyslexia (*P* = 0.013), Dysgraphia (*P* = 0.004), and Dyscalculia (*P* = 0.035), highlighting their role for reading/writing/math support. Psychological consultation correlates only with Dyslexia (*P* = 0.013) and Dysgraphia (*P* = 0.004), suggesting targeted mental health needs. Dispensatory measures (e.g., accommodations) are vital for Dyslexia (*P* = 0.001), Language Processing Disorder (*P* = 0.019), Nonverbal LD (*P* = 0.015), and Visual Motor Deficit (*P* = 0.041), while compensatory tools significantly support Dyslexia (*P* = 0.003) and Auditory Processing Disorder

**Table 5. Utilization of types of specific university services and their association with SLDs.**

| Specific Learning Disabilities | Types of Specific University Services | | | |
|---|---|---|---|---|
| | One-to-one meeting, electronic books | Psychological consultation | Use dispensatory measures | Compensatory tool |
| Dyslexia | **0.001** | **0.013** | **0.001** | **0.003** |
| Dysgraphia | **0.007** | **0.004** | 0.155 | 0.085 |
| Dyscalculia | 0.565 | **0.035** | 0.601 | 0.753 |
| Auditory processing disorder | **0.019** | 0.135 | 0.056 | **0.006** |
| Language processing disorder | **0.001** | **0.027** | **0.019** | **0.052** |
| Nonverbal learning disabilities | **0.002** | 0.589 | **0.015** | **0.030** |
| Visual perceptual/visual motor deficit | **0.017** | 0.152 | **0.041** | 0.428 |

Bolded *P* values indicate significance

(P = 0.006). Dyscalculia shows limited service correlations beyond e-books, and psychological support remains underutilized across most SLDs.

## Factors influencing the use of common services

Table 6 presents adjusted binary logistic regression results identifying predictors of frequent versus infrequent use of common university services. Among 20 demographic, academic, health, and learning disability variables analyzed, three emerged as statistically significant: (1) Students with visual perceptual/motor deficits demonstrated 3.87-fold higher odds of frequent service use (95% CI: 1.82–8.21, $P < .001$); (2) Each advancing academic year increased service engagement odds by 29% (aOR = 1.29, 95% CI: 1.09–1.53, $P = .004$); (3) Positive self-declared educational outcomes doubled service utilization likelihood (aOR = 2.32, 95% CI: 1.12–4.80, $P = .024$). Non-significant predictors included demographic factors (gender, nationality), health indicators (BMI, sleep status), and specific learning disabilities (dyscalculia: aOR = 0.95, $P = .874$; auditory processing disorder: aOR = 0.99, $P = .973$). All models report adjusted odds ratios with 95% confidence intervals (S1 Table).

## Factors influencing the use of specific services

Table 7 identifies predictors of frequent versus infrequent use of specialized support services for students with learning needs using adjusted binary logistic regression. Key significant determinants include: (1) Female gender (aOR = 2.07, 95% CI: 1.15–3.73, $P = .015$), associated with doubled odds of service engagement; (2) Sleep disturbances (aOR =

**Table 6. Factors influencing use of common services by binary regression analysis.**

| Common Services Utilization | P Value | aOR | 95% Confidence Interval for aOR | |
|---|---|---|---|---|
| | | | Lower Bound | Upper Bound |
| Age (years) | 0.546 | 1.315 | 0.541 | 3.194 |
| Gender | 0.079 | 0.593 | 0.331 | 1.062 |
| Nationality | 0.366 | 0.769 | 0.435 | 1.359 |
| Total family income (SAR) | 0.330 | 0.732 | 0.392 | 1.370 |
| Affiliated Program | 0.153 | 1.322 | 0.901 | 1.939 |
| Academic Year | **0.004** | 1.290 | 1.086 | 1.531 |
| cGPA | 0.147 | 1.376 | 0.894 | 2.117 |
| Self-declaration of educational outcome | **0.024** | 2.318 | 1.119 | 4.798 |
| Self-assessment of health | 0.060 | 0.184 | 0.031 | 1.074 |
| Chronic disease Status | 0.101 | 0.541 | 0.260 | 1.127 |
| Sleep status | 0.285 | 1.342 | 0.783 | 2.300 |
| Physical activity | 0.148 | 1.533 | 0.859 | 2.734 |
| BMI type | 0.999 | 1.000 | 0.718 | 1.393 |
| Self-declared QOL | 0.763 | 1.106 | 0.575 | 2.126 |
| Satisfaction with health | 0.623 | 0.855 | 0.459 | 1.594 |
| Dyslexia | 0.213 | 0.559 | 0.223 | 1.397 |
| Dysgraphia | 0.282 | 1.391 | 0.763 | 2.538 |
| Dyscalculia | 0.874 | 0.952 | 0.521 | 1.742 |
| Auditory processing disorder | 0.973 | 0.987 | 0.458 | 2.124 |
| Language processing disorder | 0.330 | 1.485 | 0.670 | 3.289 |
| Nonverbal learning disabilities | 0.118 | 0.536 | 0.246 | 1.170 |
| Visual perceptual/visual motor deficit | **0.000** | 3.867 | 1.822 | 8.207 |

aOR: adjusted Odds ratio; Bolded P values indicate significance.

**Table 7. Factors influencing use of specific services by binary regression analysis.**

| Status of Specific Services | P Value | aOR | 95% Confidence Interval for aOR | |
|---|---|---|---|---|
| | | | Lower Bound | Upper Bound |
| Age (years) | 0.199 | 1.822 | 0.729 | 4.550 |
| Gender | **0.015** | 2.070 | 1.150 | 3.727 |
| Nationality | 0.092 | 0.614 | 0.348 | 1.082 |
| Total family income (SAR) | 0.835 | 1.068 | 0.575 | 1.983 |
| Affiliated Program | 0.707 | 1.076 | 0.733 | 1.580 |
| Academic Year | 0.339 | 0.920 | 0.777 | 1.091 |
| cGPA | 0.092 | 1.436 | 0.942 | 2.187 |
| Self-declaration of educational outcome | 0.051 | 2.035 | 0.997 | 4.156 |
| Self-assessment of health | 0.567 | 1.621 | 0.310 | 8.473 |
| Chronic disease Status | 0.697 | 1.158 | 0.555 | 2.415 |
| Sleep status | **0.005** | 0.462 | 0.271 | 0.790 |
| Physical activity | 0.707 | 1.119 | 0.624 | 2.007 |
| BMI type | 0.334 | 0.848 | 0.608 | 1.184 |
| Self-declared QOL | 0.420 | 1.306 | 0.683 | 2.497 |
| Satisfaction with health | 0.298 | 0.717 | 0.383 | 1.342 |
| Dyslexia | **0.036** | 2.728 | 1.067 | 6.976 |
| Dysgraphia | 0.273 | 1.403 | 0.766 | 2.570 |
| Dyscalculia | 0.290 | 0.724 | 0.399 | 1.315 |
| Auditory processing disorder | **0.018** | 2.521 | 1.174 | 5.411 |
| Language processing disorder | 0.633 | 0.829 | 0.384 | 1.790 |
| Nonverbal learning disabilities | 0.536 | 0.786 | 0.368 | 1.682 |
| Visual perceptual/visual motor deficit | 0.365 | 0.722 | 0.356 | 1.461 |

aOR: adjusted Odds ratio; Bolded *P* values indicate significance.

0.46, 95% CI: 0.27–0.79, *P* = .005), linked to 54% lower utilization; and (3) Specific learning disabilities—dyslexia (aOR = 2.73, 95% CI: 1.07–6.98, *P* = .036) and auditory processing disorders (aOR = 2.52, 95% CI: 1.17–5.41, *P* = .018)—both showing >2.5× higher service use. Academic achievement (cGPA: aOR = 1.44, *P* = .092) and educational self-perception (aOR = 2.04, *P* = .051) approached significance. Non-significant factors encompassed nationality, chronic health conditions, and other learning disabilities (e.g., dyscalculia: aOR = 0.72, *P* = .290). Results report adjusted odds ratios with 95% confidence intervals (S1 Table).

## Correlation analysis of specific learning disabilities with outcome variables

Table 8 presents the Pearson correlation analysis for specific learning disabilities and outcome variables. The analysis shows significant correlations between various specific learning disabilities and the use of services. Dyslexia shows a significant positive correlation with the use of specific services (r = 0.193, *P* = 0.001), but not with common services (r = 0.095, *P* = 0.099). Dysgraphia is significantly correlated with common services (r = 0.144, *P* = 0.012) and also has positive correlations with other learning disabilities. Dyscalculia shows significant correlations with other learning disabilities, but not with the use of common or specific services. Auditory processing disorder (r = 0.187, *P* = 0.001) and language processing disorders (r = 0.162, *P* = 0.005 for common services; r = 0.514, *P* < 0.001 for specific services) both show significant correlations with the use of services. Additionally, visual perceptual/visual motor deficit shows a significant positive correlation with common services (r = 0.252, *P* < 0.001), but not with specific services. The correlations indicate significant relationships

**Table 8.  Pearson correlation analysis for specific learning disabilities and outcome variables.**

| | | Specific Services | Common services | Dyslexia | Dysgraphia | Dyscalculia | Auditory processing disorder | Language processing disorder | Nonverbal learning disabilities | Visual perceptual/ visual motor deficit |
|---|---|---|---|---|---|---|---|---|---|---|
| Specific Services | Pearson Correlation | 1 | 0.095 | 0.193** | 0.111 | 0.063 | 0.187** | 0.115* | 0.090 | 0.070 |
| | Sig. (2-tailed) | | 0.099 | 0.001 | 0.054 | 0.275 | 0.001 | 0.046 | 0.117 | 0.226 |
| Common services | Pearson Correlation | 0.095 | 1 | 0.085 | 0.144* | 0.082 | 0.081 | 0.162** | 0.095 | 0.252** |
| | Sig. (2-tailed) | 0.099 | | 0.143 | 0.012 | 0.154 | 0.160 | 0.005 | 0.100 | 0.000 |
| Dyslexia | Pearson Correlation | .193** | 0.085 | 1 | 0.267** | 0.301** | 0.282** | 0.419** | 0.316** | 0.370** |
| | Sig. (2-tailed) | 0.001 | 0.143 | | 0.000 | 0.000 | 0.000 | 0.000 | 0.000 | 0.000 |
| Dysgraphia | Pearson Correlation | 0.111 | 0.144* | 0.267** | 1 | 0.290** | 0.224** | 0.410** | 0.395** | 0.338** |
| | Sig. (2-tailed) | 0.054 | 0.012 | 0.000 | | 0.000 | 0.000 | 0.000 | 0.000 | 0.000 |
| Dyscalculia | Pearson Correlation | 0.063 | 0.082 | 0.301** | 0.290** | 1 | 0.330** | 0.419** | 0.289** | 0.214** |
| | Sig. (2-tailed) | 0.275 | 0.154 | 0.000 | 0.000 | | 0.000 | 0.000 | 0.000 | 0.000 |
| Auditory processing disorder | Pearson Correlation | 0.187** | 0.081 | 0.282** | 0.224** | 0.330** | 1 | 0.514** | 0.367** | 0.255** |
| | Sig. (2-tailed) | 0.001 | 0.160 | 0.000 | 0.000 | 0.000 | | 0.000 | 0.000 | 0.000 |
| Language processing disorder | Pearson Correlation | 0.115* | 0.162** | 0.419** | 0.410** | 0.419** | 0.514** | 1 | 0.540** | 0.453** |
| | Sig. (2-tailed) | 0.046 | 0.005 | 0.000 | 0.000 | 0.000 | 0.000 | | 0.000 | 0.000 |
| Nonverbal learning disabilities | Pearson Correlation | 0.090 | 0.095 | 0.316** | 0.395** | 0.289** | 0.367** | 0.540** | 1 | 0.476** |
| | Sig. (2-tailed) | 0.117 | 0.100 | 0.000 | 0.000 | 0.000 | 0.000 | 0.000 | | 0.000 |
| Visual perceptual/ visual motor deficit | Pearson Correlation | 0.070 | 0.252** | 0.370** | 0.338** | 0.214** | 0.255** | 0.453** | 0.476** | 1 |
| | Sig. (2-tailed) | 0.226 | 0.000 | 0.000 | 0.000 | 0.000 | 0.000 | 0.000 | 0.000 | |

**. Correlation is significant at the 0.01 level (2-tailed).

*. Correlation is significant at the 0.05 level (2-tailed).

between learning disabilities and the use of services, particularly with language processing disorder and visual perceptual/ visual motor deficit affecting common service use.

## Discussion

The use of university services by students with specific learning disabilities (SLDs) has been a critical area of research due to its implications for academic success and student well-being. This study explored the relationship between students' SLD status and their utilization of both common and specific university services. The results of this study suggest a significant association between the presence of SLDs and the underutilization of both common and specific support services. These findings align with existing literature that indicates barriers to service use among students with learning disabilities, including lack of awareness, stigma, and insufficient accommodations [19].

The data indicate that while a majority (55.3%) of students reported infrequent usage of common university services, academic-related resources such as lectures (75%), Learning Management Systems (LMS) (74.4%), and library services (67.6%) remain central to student engagement. This preference aligns with previous studies indicating that students with learning difficulties often prioritize core instructional resources that directly impact their academic performance [20]. Notably, support and wellness services, particularly psychological counseling (31%) and fitness/sports facilities (37.4%), are significantly underutilized, despite their documented importance in promoting student well-being and academic persistence [21]. This finding reflects a well-documented trend in the literature, where students with disabilities often underuse mental

health services due to stigma, lack of awareness, or limited-service accessibility [22]. Analysis of specific university services revealed that one-to-one meetings and electronic books (63.2%) were among the most utilized resources. These findings demonstrate the value of personalized and flexible academic support for students with SLDs, echoing earlier research emphasizing the effectiveness of tailored instruction and assistive technologies in higher education [23].

The study's results indicate that students with specific learning disabilities such as dyslexia, dysgraphia, auditory processing disorder, language processing disorder, and visual perceptual/visual motor deficits were less likely to use common university services compared to students without these disabilities. This finding is noteworthy because these services—such as academic advising, psychological assistance, and the use of digital platforms like Moodle and academic databases—can significantly contribute to student success by providing essential academic support and mental health resources [24,25].

For instance, students with dyslexia were found to use common services like the library and academic portals less frequently, a result that reflects the broader challenges faced by students with learning disabilities in accessing resources. As reported by Hebert et al [26], students with dyslexia often experience difficulties in reading and processing written information, which could make engagement with traditional academic services, such as the library or online platforms, more challenging. Furthermore, this underutilization may be influenced by the lack of tailored support to help them navigate these services effectively.

Similarly, students with auditory processing and language processing disorders also demonstrated lower usage of common services. This underuse may be tied to difficulties in processing verbal information or understanding academic content, which in turn could lead to students feeling overwhelmed or disheartened when attempting to access academic resources [27]. These findings emphasize the need for universities to enhance accessibility to common services for students with learning disabilities, offering modifications such as additional support in accessing online platforms or assistance with reading and comprehension in library environments.

Interestingly, visual perceptual/visual motor deficits were associated with particularly low usage of common services like the library and Moodle. Visual perceptual difficulties can make tasks such as reading, navigating websites, or engaging in classroom activities particularly strenuous [28]. This underscores the need for greater accommodations, such as providing materials in formats compatible with assistive technologies or enhancing the design of online platforms to be more accessible for students with visual impairments.

The use of specific university services, such as academic support (e.g., one-on-one tutoring, electronic books), psychological support, and compensatory measures (e.g., extended exam time, split lecture material), was also significantly influenced by the presence of learning disabilities. As expected, students with learning disabilities were more likely to report using specific services such as one-on-one tutoring and compensatory measures, which are designed to assist them in overcoming the barriers posed by their disabilities. However, the frequency of use of these services was still lower than expected, considering the academic challenges these students face.

Despite the availability of resources tailored to their needs—such as digital reading tools and extended exam times—students with dyslexia in this study were less likely to take advantage of these accommodations. This aligns with the findings of another study [29], which noted that students with dyslexia often underutilize available academic support, possibly due to a lack of awareness about the accommodations or fear of stigma related to their condition. The underutilization of specific services by students with dyslexia is concerning, as these services are designed to help them succeed academically by providing essential resources like additional time, alternative formats for reading, or one-on-one tutoring sessions.

A similar pattern emerged for students with auditory processing disorder and language processing disorder, who also reported lower usage of specific services despite their eligibility for academic support. The reasons behind this underuse are multifaceted but likely include a combination of psychological barriers, such as feelings of inadequacy or fear of being singled out, and institutional factors, including a lack of targeted outreach or personalized support. Studies have shown that students with auditory and language processing disorders often face significant challenges in the traditional classroom setting, making specific academic support services essential to their success [30]. Yet, despite the availability of

these services, students with these conditions may hesitate to engage with them, indicating a critical gap in the accessibility of the support systems in place.

Visual perceptual/visual motor deficits also showed a notable impact on the use of specific services. Students with this condition reported using services like electronic books or compensatory measures, such as extended exam time significantly less than their peers without visual deficits. This finding highlights the fact that even when specific services are available, students with learning disabilities may struggle to access or fully utilize them without proper guidance or advocacy [31]. It is essential that universities not only provide these services but also ensure that students are equipped with the knowledge and confidence to take full advantage of them. This can be achieved by offering targeted support, such as one-on-one consultations or training sessions on how to use assistive technologies effectively.

The Binary regression analysis presented several significant predictors of service usage, particularly for specific services. Gender differences were observed, with female students utilizing specific services more frequently than male students. This aligns with previous studies that suggest women are more likely to seek help when faced with academic challenges [32], whereas men may be less inclined to use support services due to societal expectations of self-reliance. This gender disparity in help-seeking behavior is especially pertinent for students with learning disabilities, who may already feel marginalized or reluctant to seek additional help due to the stigma associated with their condition.

Sleep status also played a significant role in service usage, particularly for specific services. Students with below-normal sleep patterns were more likely to use specific services, such as psychological support and compensatory measures. This is consistent with research indicating that sleep deprivation has a profound effect on cognitive functioning and academic performance [33]. Sleep deprivation is a common issue among university students, particularly those with learning disabilities, who may struggle with time management and academic demands. This suggests that improving students' sleep habits could have a positive impact on their academic engagement and use of support services.

The presence of specific learning disabilities, such as dyslexia and auditory processing disorder was a strong predictor of the use of specific services. This highlights the importance of universities recognizing the unique needs of students with these conditions and providing specialized resources to support them. While the need for these services is clear, the underutilization of these services by students with learning disabilities points to systemic issues that need to be addressed. These include increasing awareness of available services, reducing the stigma associated with learning disabilities, and ensuring that academic support services are accessible, inclusive, and well-publicized.

## Strengths of the study

This study offers several key strengths. It provides a comprehensive and data-driven analysis of university service utilization among students with specific learning disabilities (SLDs), using both frequency metrics and statistical correlations (Pearson Chi-Square and binary regression). The inclusion of a wide spectrum of SLDs—such as dyslexia, dysgraphia, auditory and language processing disorders, and visual perceptual/motor deficits—adds depth and nuance to the findings. Another notable strength is the dual focus on both common and specialized university services, allowing for a holistic understanding of how students with SLDs navigate academic and wellness resources. The study also integrates important demographic and behavioral predictors such as gender and sleep status, which enriches the contextual relevance of the results and aligns them with prior empirical research. Overall, the findings contribute meaningful insights to the discourse on inclusive education and support service optimization in higher education settings.

## Limitations of the study

While the findings provide valuable insights into the use of university services among students with specific learning disabilities, several limitations should be considered. The cross-sectional design captures valuable snapshots of service usage patterns but inherently limits causal inference. While we identified significant correlations between student

characteristics and service utilization, we cannot determine whether these factors *precede* engagement (e.g., whether distress drives help-seeking) or result from it (e.g., whether services alleviate distress). Longitudinal tracking of student cohorts is needed to establish temporal precedence and identify causal drivers of long-term engagement with university support systems. Furthermore, the study relied on self-reported data, which is subject to potential biases including social desirability, recall inaccuracies, and—specifically for learning disabilities—discrepancies between self-identification and clinical diagnoses. Diagnostic standards for SLDs vary geographically, and our inclusive approach (accepting self-reported symptoms without clinical verification) may have captured students with undiagnosed challenges while potentially including others whose experiences may not align with formal diagnostic criteria. Future research should triangulate self-reports with institutional diagnostic records where available, while also investigating contextual factors influencing students' understanding and reporting of learning disabilities.

Moreover, the study was conducted within a single university, which limits the generalizability of the findings. Different universities may offer different types of services or have distinct support structures in place, which could influence service usage. Further studies should consider a broader sample of students from multiple universities, both within and outside of Saudi Arabia, to provide a more comprehensive understanding of the factors influencing service utilization.

Lastly, despite employing random selection from institutional enrollment lists, our 17.76% participation rate (302/1,700) introduces potential non-response bias. While demographic comparisons confirmed broad representativeness across academic programs, voluntary participation likely overrepresented students with strong opinions about services or prior service exposure. This may underestimate barriers faced by reluctant help-seekers, particularly underrepresented students. Generalizability should thus be tempered by acknowledging possible under-coverage of disengaged populations.

## Recommendations

Given the findings of this study, it is recommended that universities increase outreach efforts to ensure students with learning disabilities are aware of and able to access the support services available to them. Universities should focus on reducing stigma, increasing self-advocacy skills among students with SLDs, and providing more tailored services that are responsive to the specific needs of these students. Additionally, increasing the visibility and accessibility of assistive technologies and compensatory measures is critical for ensuring that all students, regardless of their learning difficulties, have equal access to educational opportunities.

## Conclusion

This study highlights the critical need for accessible, inclusive, and well-publicized support services tailored to students with specific learning disabilities (SLDs). The findings reveal a significant association between the presence of SLDs and the underutilization of both common university resources—such as academic advising, library services, and wellness facilities—and specific support services, including psychological counseling and compensatory accommodations. Despite the availability of these resources, barriers such as lack of awareness, stigma, and limited accessibility continue to hinder effective service use. The study also highlights key predictors of service utilization, including gender and sleep status, offering important insights for targeted interventions. To foster equity and academic success, universities must not only enhance the visibility and accessibility of support services but also implement inclusive strategies that actively encourage and empower students with SLDs to engage with them. Creating a supportive academic environment will ensure that all students—regardless of their learning needs—can fully participate, thrive, and succeed in higher education.

## Acknowledgment

The authors extend their appreciation to the King Salman Center for Disability Research for funding this work through Research Group no KSRG-2024-027.

# Supporting information

**S1 Table. Factors influencing the use of common and specific services.**
(PDF)

**S1 File. Study Questionnaire.**
(PDF)

**S2 File. Raw data of the study.**
(XLSX)

# Author contributions

**Conceptualization:** Syed Mohammed Basheeruddin Asdaq, Mamdouh Saleh Alharbi, Mohanad Abdullah Alansari, Saleh Alshuqayr, Malek Fares Alanazi, Saleh Rajih Alshahrani, Naira Nayeem, Walaa F. Alsanie, Abdulhakeem S. Alamri, Majid Alhomrani, Rafiulla Gilkaramenthi.

**Data curation:** Syed Mohammed Basheeruddin Asdaq, Mohanad Abdullah Alansari, Saleh Alshuqayr, Amal F. Alshammary.

**Formal analysis:** Syed Mohammed Basheeruddin Asdaq, Saleh Rajih Alshahrani.

**Funding acquisition:** Syed Mohammed Basheeruddin Asdaq.

**Investigation:** Mamdouh Saleh Alharbi, Mohanad Abdullah Alansari, Saleh Alshuqayr, Malek Fares Alanazi, Abdulhakeem S. Alamri, Rafiulla Gilkaramenthi.

**Methodology:** Mamdouh Saleh Alharbi, Naira Nayeem, Amal F. Alshammary, Rafiulla Gilkaramenthi.

**Project administration:** Syed Mohammed Basheeruddin Asdaq.

**Resources:** Saleh Rajih Alshahrani, Walaa F. Alsanie, Majid Alhomrani.

**Software:** Walaa F. Alsanie, Majid Alhomrani.

**Supervision:** Syed Mohammed Basheeruddin Asdaq.

**Visualization:** Saleh Alshuqayr.

**Writing – original draft:** Mamdouh Saleh Alharbi, Mohanad Abdullah Alansari, Saleh Alshuqayr, Malek Fares Alanazi, Saleh Rajih Alshahrani, Walaa F. Alsanie, Abdulhakeem S. Alamri, Majid Alhomrani, Amal F. Alshammary, Rafiulla Gilkaramenthi.

**Writing – review & editing:** Syed Mohammed Basheeruddin Asdaq, Naira Nayeem.

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
