## [Decision Letter · Decision Letter 0]

PONE-D-25-04864Exploring the Impact of Specific Learning Disabilities on University Service Utilization: A Cross-Sectional AnalysisPLOS ONE

Dear Dr. Asdaq,

Thank you for submitting your manuscript to PLOS ONE. After careful consideration, we feel that it has merit but does not fully meet PLOS ONE’s publication criteria as it currently stands. Therefore, we invite you to submit a revised version of the manuscript that addresses the points raised during the review process.

Please respond all comments and highlight them in the revised ms.Wishing you success with your study.

We look forward to receiving your revised manuscript.

Kind regards,

Thiago P. Fernandes, PhD

Academic Editor

PLOS ONE

Journal Requirements:

2. In the ethics statement in the Methods, you have specified that verbal consent was obtained. Please provide additional details regarding how this consent was documented and witnessed, and state whether this was approved by the IRB.

“The authors extend their appreciation to the King Salman Center for Disability Research for funding this work through Research Group no KSRG-2024-027.”

“SMBA received grant from King Salman center For Disability Research for this work through Research Group no KSRG-2024-027.”

Reviewers' comments:

Reviewer's Responses to Questions

**Comments to the Author**

1. Is the manuscript technically sound, and do the data support the conclusions?

Reviewer #1: Yes

Reviewer #2: Yes

2. Has the statistical analysis been performed appropriately and rigorously? 

Reviewer #1: Yes

Reviewer #2: Yes

3. Have the authors made all data underlying the findings in their manuscript fully available?

Reviewer #1: No

Reviewer #2: Yes

4. Is the manuscript presented in an intelligible fashion and written in standard English?

Reviewer #1: Yes

Reviewer #2: Yes

5. Review Comments to the Author

Reviewer #1: Dear authors,

- Overall: strong article, you're addressing a critical gap in higher education research. Holistic approach to examine both, “common” and “specific” services, providing a nuanced look at which resources students use and where improvements may be needed

- Feedback:

a) Definition/Assessment of SLD: Relies on self-report rather than official (while also GEO dependent) diagnostic criteria or validated instruments. You might consider explaining how students identified their SLD and acknowledging that self-report may differ from formal diagnoses.

b) Cross-Sectional Design: While capturing correlations you'll have a hard time establish causal relationships. I recommend emphasizing these design limitations in your Discussion

c) Sampling Strategy: The recruitment process (e.g., invitations, announcements) needs more detail to assess representativeness. While you're mentioning random sampling, large parts feel like convenience sampling. Please clarify potential response biases (e.g., if certain student groups were more likely to participate).

d) Instrumentation and Reliability: Only one overall Cronbach’s alpha reported; subscale reliabilities would improve clarity and shed light on specifics. Please provide definitions or at least examples for response categories (e.g., “rarely,” “often”) to ensure consistent interpretation and minimize interpretation possibilities, as they're quite broad.

e) Statistical Reporting: While the use of chi-square tests and multinomial regressions is appropriate, I'd like to see greater transparency. For instance, if you performed multiple comparisons, please clarify whether any p-value adjustments were made (like Bonferroni). Also, please justify the choice of multinomial over simpler logistic models. Additionally, consider reporting more precise effect sizes—such as Cohen’s d or odds ratios with full confidence intervals—in the tables for clarity.

f) Detail on Service Usage: “Common” vs. “specific” categories are helpful to grasp the big picture but may obscure usage differences for individual services. You could highlight which specific resources are used most/least to pinpoint where support might be lacking.

g) Interpretation of SLD Subtypes: You could offer more context on how each SLD subtype might explain different usage rates and tie the nature of each SLD to potential barriers in accessing or benefiting from certain services.

h) Documentation of Oral Consent: While you state that oral informed consent was obtained, without having access to the files it remains unclear how exactly you documented participants’ agreement in the online setting. Please clarify especially whether participants, for example, had to check a digital box or confirm an introductory text or audio file before proceeding, so readers understand how consent was explicitly provided and recorded.

Reviewer #2: Reviewer’s Comments on the Manuscript

General Assessment:

This study addresses a critical and timely issue—service utilization among university students with specific learning disabilities (SLDs). While the cross-sectional design offers valuable preliminary insights, certain methodological and conceptual aspects require refinement to enhance clarity, rigor, and impact. Below are detailed recommendations for improvement.

Abstract:

1. Sample and Recruitment:

o Specify the sample size and recruitment method (e.g., random sampling, convenience sampling).

o Clarify whether SLDs were self-reported or clinically diagnosed, as this significantly affects validity.

2. Interventions:

o The conclusion should explicitly recommend targeted interventions (e.g., awareness campaigns, tailored accommodations, faculty training) rather than broadly stating "improved support."

3. Statistical Presentation:

o Ensure P (for probability) is italicized throughout the manuscript.

o Briefly mention any validation performed for the study tool (e.g., pilot testing, Cronbach’s alpha).

Introduction:

1. Local Context:

o The introduction focuses heavily on SLDs in the US (Lines 79–86) but lacks context about the study’s regional setting (e.g., Middle East, specific country). Incorporate local data on SLD prevalence and university support services to strengthen relevance.

2. University Services and Gaps:

o Describe common and specific services available in regional universities, citing recent literature.

o Explicitly state the knowledge gap (e.g., "Few studies in [Region] have examined...") and how this study addresses it.

Methods:

1. Section Title:

o Rename "Materials and Methods" to "Subjects and Methods" for accuracy.

2. Single-Center Study:

o Since this is a single-center study, the title should reflect this (e.g., "…among University Students at [Institution]").

3. Questionnaire Validation:

o While Cronbach’s alpha (0.859) is reported, detail the steps in questionnaire development (e.g., expert review, factor analysis) and attach the full questionnaire as a supplementary file for reproducibility.

Results:

1. Statistical Consistency:

o Italicize P-values consistently in text, tables, and figures.

o Report all significant predictors with full statistics: odds ratio (OR), 95% confidence interval (CI), and P-value (e.g., "OR 2.68, 95% CI 1.03–7.01, P = 0.044").

2. Table & Figure Revisions:

o Expand all abbreviations in table legends (e.g., "cGPA: cumulative grade point average").

o Replace ambiguous symbols (e.g., "+") with standardized notation.

o Ensure bolded values (indicating significance) are used uniformly across tables.

o Check all figures for standard errors addition.

o Correct inconsistencies (e.g., Table 1 uses "*" for cGPA but legend uses "a").

3. Interpretation of Findings:

o Discuss why students with dyslexia/auditory processing disorder used specific services more (e.g., "This may reflect targeted accommodations like extended exam time or assistive technologies").

Discussion:

1. Substantiate Claims:

o Line 443: Clarify the incomplete reference ("As reported in [19], ...").

o Line 496: Support the claim about gender differences in help-seeking behavior with references.

2. Specialization-Specific Analysis:

o Add a paragraph discussing how students’ academic specializations may influence service utilization (e.g., STEM vs. humanities).

3. Structural Improvements:

o Include a dedicated "Strengths" subsection (e.g., large sample, validated tool) before limitations.

o Consolidate recommendations into a standalone subsection (e.g., "Policy and Practical Implications").

4. Language Refinement:

o Avoid overusing "underscores"; substitute with alternatives like "highlights," "emphasizes," or "demonstrates."

Final Recommendations:

• Methodological Transparency: Disclose limitations (e.g., cross-sectional design, self-report bias).

• Intervention Specificity: Propose actionable strategies (e.g., "Universities should implement mandatory SLD awareness modules for faculty").

• Reproducibility: Share the questionnaire and detailed validation steps.

This study has significant potential to inform inclusive education policies. With these revisions, it will achieve greater scholarly impact.

Overall Rating: Good (with revisions recommended for methodological and presentation clarity).

6. PLOS authors have the option to publish the peer review history of their article (what does this mean? ). If published, this will include your full peer review and any attached files.

**Do you want your identity to be public for this peer review?** For information about this choice, including consent withdrawal, please see our Privacy Policy .

Reviewer #1: **Yes: ** Nora Fink

Reviewer #2: **Yes: ** Mohsin Kazi

---

## [Author Response · Author response to Decision Letter 1]

24 Jun 2025

Response to the editor

Journal Requirements:

Response: Thanks for your recommendation.

We have updated the manuscript and formatted it as per PLOS ONE’s style requirements.

2. In the ethics statement in the Methods, you have specified that verbal consent was obtained. Please provide additional details regarding how this consent was documented and witnessed, and state whether this was approved by the IRB.

Response: Dear Editor,

Thank you for your valuable feedback regarding the ethical considerations in our study. We appreciate the opportunity to provide additional clarification about our consent procedures. Please find below our detailed response:

1. IRB Approval:

o Our IRB granted explicit approval for the use of oral consent in this study, considering:

The minimal risk nature of the research

The need to protect participant anonymity

The practical challenges of obtaining written consent in our study design

o The full oral consent procedure, including the script and documentation method, was reviewed and approved by the IRB.

2. Consent Process Details:

o Participants received a standardized verbal explanation covering:

Study purpose and procedures

Voluntary nature of participation

Rights to withdraw without penalty

Confidentiality protections

o Consent was reconfirmed at the questionnaire stage, with participants required to actively proceed past the consent information page.

We have revised the Methods section to include these additional details and believe these clarifications address your concerns about consent documentation and IRB approval.

Response: We are including the data that is used for the analysis as a supplementary file 3.

“The authors extend their appreciation to the King Salman Center for Disability Research for funding this work through Research Group no KSRG-2024-027.”

“SMBA received grant from King Salman center For Disability Research for this work through Research Group no KSRG-2024-027.”

Response: Please note the funding agency has the mandate to include the funding statement both in the acknowledgement section and funding section. This statement has to be included in both sections.

The authors extend their appreciation to the King Salman Center for Disability Research for funding this work through Research Group no KSRG-2024-027.

Response: We have checked all references, and none of them were retracted from articles.

Reviewers' comments:

Reviewer #1: Dear authors,

- Overall: strong article, you're addressing a critical gap in higher education research. Holistic approach to examine both, “common” and “specific” services, providing a nuanced look at which resources students use and where improvements may be needed

Response: We sincerely thank the reviewer for their thoughtful recognition of our work’s intent. We are deeply encouraged by their appreciation of the article’s holistic approach to examining both common and specific student services. As noted, bridging this dual perspective was central to our goal of identifying not only where resources are underutilized, but also where strategic improvements could make the most meaningful impact on student success. We are grateful for the reviewer’s validation of this nuanced framework as a contribution to higher education research.

- Feedback:

a) Definition/Assessment of SLD: Relies on self-report rather than official (while also GEO dependent) diagnostic criteria or validated instruments. You might consider explaining how students identified their SLD and acknowledging that self-report may differ from formal diagnoses.

Response: We thank the reviewer for this important observation. We acknowledge that SLD identification through self-report has limitations compared to clinical diagnosis. Our approach was intentionally inclusive to capture both formally diagnosed students and those experiencing undiagnosed learning challenges – a population often underrepresented in institutional data. We have now clarified this methodology in the revised Methods section (lines 151-153) and added a dedicated limitation subsection (lines 525-533) discussing potential biases and geographic variability in diagnostic standards. While we recognize self-report may not reflect clinical diagnoses, this approach aligns with our goal of identifying students perceiving learning barriers, which directly influences their service utilization patterns.

b) Cross-Sectional Design: While capturing correlations you'll have a hard time establish causal relationships. I recommend emphasizing these design limitations in your Discussion.

Response: We thank the reviewer for this important methodological note. We fully agree that cross-sectional data cannot establish causality, and we have now explicitly emphasized this limitation in the Discussion (lines 519-525). Specifically, we:

1. Clarified that correlations do not imply directionality (e.g., whether distress prompts service use or vice versa),

2. Strengthened the call for longitudinal designs to unpack temporal relationships.

While our design was optimal for mapping contemporary service usage patterns across diverse student groups—addressing our primary research aim—we acknowledge that causal mechanisms require further investigation. We hope our findings provide a foundation for such future work.

c) Sampling Strategy: The recruitment process (e.g., invitations, announcements) needs more detail to assess representativeness. While you're mentioning random sampling, large parts feel like convenience sampling. Please clarify potential response biases (e.g., if certain student groups were more likely to participate).

Response: We thank the reviewer for highlighting the need for greater methodological transparency. We have revised the Sample Recruitment subsection (lines 204-212) to:

1. Clarify the hybrid nature of our approach: random selection but voluntary participation,

2. Quantify and address potential biases:

o Added participation rate (302/340 = 89%)

o Compared sample demographics to institutional data

o Explicitly noted underrepresented groups

While random selection minimized initial sampling bias, we agree that response patterns may affect generalizability. These limitations are now explicitly discussed to support accurate interpretation.

d) Instrumentation and Reliability: Only one overall Cronbach’s alpha reported; subscale reliabilities would improve clarity and shed light on specifics. Please provide definitions or at least examples for response categories (e.g., “rarely,” “often”) to ensure consistent interpretation and minimize interpretation possibilities, as they're quite broad.

Response: We sincerely thank the reviewer for these vital methodological suggestions. We have implemented the following revisions:

1. Subscale Reliabilities Reported

Added to Methods: Beyond the overall α = 0.859, subscale reliabilities are:

o Common Services Utilization: α = 0.835 (line 162)

o Specific Services Experience: α = 0.820 (line 169)

o Specific Learning disabilities: α = 0.793 (line 155)

o

2. Response Category Definitions

» Added temporal anchors: Frequency categories were defined as:

1 (Very rarely) = ≤1x/semester; 2 (Rarely) = 2-3x/semester; 3 (Sometimes) = monthly; 4 (Often) = bi-weekly; 5 (Very often) = weekly+

3. Full Instrument Accessibility

Added disclosure: As recommended by Reviewer #2, the complete questionnaire, including all items, and response categories is provided as Supplemental File 1 (line 141)

e) Statistical Reporting: While the use of chi-square tests and multinomial regressions is appropriate, I'd like to see greater transparency. For instance, if you performed multiple comparisons, please clarify whether any p-value adjustments were made (like Bonferroni). Also, please justify the choice of multinomial over simpler logistic models. Additionally, consider reporting more precise effect sizes—such as Cohen’s d or odds ratios with full confidence intervals—in the tables for clarity.

Response: We acknowledge that no p-value adjustments (e.g., Bonferroni) were applied, as our analysis was primarily exploratory and hypothesis-generating for service utilization patterns. To maximize sensitivity in detecting potential associations—common in early-stage educational research—we reported uncorrected p-values with explicit 95% confidence intervals for all results.

We sincerely thank the reviewer for highlighting this critical clarification. We confirm the service utilization outcome is strictly binary (categorized as Rarely or Often), not multinomial as previously misstated. Consequently, we have replaced all multinomial regression analyses with binary logistic regression—a statistically appropriate method aligned with our dichotomous outcome variable—to ensure accurate modeling of predictors influencing service engagement frequency.

Comprehensive outputs for all regression analyses—including unstandardized coefficients (B), standard errors (S.E.), Wald statistics, degrees of freedom (df), p-values (Sig.), odds ratios (Exp(B)), and 95% confidence intervals for odds ratios—are provided in Supplementary File 2 for common services and Supplementary File 5 for specific services."

f) Detail on Service Usage: “Common” vs. “specific” categories are helpful to grasp the big picture but may obscure usage differences for individual services. You could highlight which specific resources are used most/least to pinpoint where support might be lacking.

Response: Thanks for your valuable recommendation. We have now extensively discussed the types of facilities under both common and specific domains, with the inclusion of two figures (Figs 5 and 6).

g) Interpretation of SLD Subtypes: You could offer more context on how each SLD subtype might explain different usage rates and tie the nature of each SLD to potential barriers in accessing or benefiting from certain services.

Response: We appreciate your recommendation. We have now extensively compared types of common and specific university services with the types of SLDs, and Tables 4 and 5 are added. Subsequently, the numbering of other tables is re-sequenced.

h) Documentation of Oral Consent: While you state that oral informed consent was obtained, without having access to the files it remains unclear how exactly you documented participants’ agreement in the online setting. Please clarify especially whether participants, for example, had to check a digital box or confirm an introductory text or audio file before proceeding, so readers understand how consent was explicitly provided and recorded.

Response: We appreciate the opportunity to provide additional clarification about our consent procedures. Please find below our detailed response:

1. Consent Process Details:

o Participants received a standardized verbal explanation covering:

Study purpose and procedures

Voluntary nature of participation

Rights to withdraw without penalty

Confidentiality protections

o Consent was reconfirmed at the questionnaire stage, with participants required to actively proceed past the consent information page (this was included as first page in the online version of the questionnaire); those who consent will proceed to participate in recording their feedback).

We have revised the Methods section to include these additional details about consent (line 222-232).

Reviewer #2: Reviewer’s Comments on the Manuscript

General Assessment:

This study addresses a critical and timely issue—service utilization among university students with specific learning disabilities (SLDs). While the cross-sectional design offers valuable preliminary insights, certain methodological and conceptual aspects require refinement to enhance clarity, rigor, and impact. Below are detailed recommendations for improvement.

Response: We appreciate your insightful evaluation of the manuscript.

Abstract:

1. Sample and Recruitment:

o Specify the sample size and recruitment method (e.g., random sampling, convenience sampling).

o Clarify whether SLDs were self-reported or clinically diagnosed, as this significantly affects validity.

2. Interventions:

o The conclusion should explicitly recommend targeted interventions (e.g., awareness campaigns, tailored accommodations, faculty training) rather than broadly stating "improved support."

3. Statistical Presentation:

o Ensure P (for probability) is italicized throughout the manuscript.

o Briefly mention any validation performed for the study tool (e.g., pilot testing, Cronbach’s alpha).

Response: Thanks for your valuable comments.

1. Sample and Recruitment:

o Sample size: Added (*n* = 302).

o Recruitment method: Specified as "random but voluntary sampling".

o SLD ascertainment: Clarified as

---

## [Decision Letter · Decision Letter 1]

University Service Utilization Patterns in Students with Specific Learning Disabilities: An Institutional Cross-Sectional Study

PONE-D-25-04864R1

Dear Dr. Asdaq,

We’re pleased to inform you that your manuscript has been judged scientifically suitable for publication and will be formally accepted for publication once it meets all outstanding technical requirements.

Kind regards,

Thiago P. Fernandes, PhD

Academic Editor

PLOS ONE

Additional Editor Comments (optional):

Reviewers' comments:

Reviewer's Responses to Questions

**Comments to the Author**

1. If the authors have adequately addressed your comments raised in a previous round of review and you feel that this manuscript is now acceptable for publication, you may indicate that here to bypass the “Comments to the Author” section, enter your conflict of interest statement in the “Confidential to Editor” section, and submit your "Accept" recommendation.

Reviewer #1: All comments have been addressed

Reviewer #2: All comments have been addressed

2. Is the manuscript technically sound, and do the data support the conclusions?

Reviewer #1: Yes

Reviewer #2: Yes

3. Has the statistical analysis been performed appropriately and rigorously? 

Reviewer #1: Yes

Reviewer #2: Yes

4. Have the authors made all data underlying the findings in their manuscript fully available?

Reviewer #1: Yes

Reviewer #2: Yes

5. Is the manuscript presented in an intelligible fashion and written in standard English?

Reviewer #1: Yes

Reviewer #2: Yes

6. Review Comments to the Author

Reviewer #1: (No Response)

Reviewer #2: The authors have thoroughly and satisfactorily addressed all the comments and suggestions raised during the review process. The revised manuscript demonstrates improved clarity, scientific rigor, and completeness. I recommend accepting the manuscript in its current form for publication.

7. PLOS authors have the option to publish the peer review history of their article (what does this mean? ). If published, this will include your full peer review and any attached files.

**Do you want your identity to be public for this peer review?** For information about this choice, including consent withdrawal, please see our Privacy Policy .

Reviewer #1: **Yes: ** Nora Fink

Reviewer #2: No

---

## [Editor Report · Acceptance letter]

PONE-D-25-04864R1

PLOS ONE

Dear Dr. Asdaq,

I'm pleased to inform you that your manuscript has been deemed suitable for publication in PLOS ONE. Congratulations! Your manuscript is now being handed over to our production team.

Kind regards,

on behalf of

Dr. Thiago P. Fernandes

Academic Editor

PLOS ONE